# Controversy in the Use of CD38 Antibody for Treatment of Myeloma: Is High CD38 Expression Good or Bad?

**DOI:** 10.3390/cells9020378

**Published:** 2020-02-06

**Authors:** Torben Plesner, Niels W. C. J. van de Donk, Paul G. Richardson

**Affiliations:** 1Vejle Hospital and University of Southern Denmark, 7100 Vejle, Denmark; 2Amsterdam UMC, Vrije Universiteit Amsterdam, Department of Hematology, Cancer Center Amsterdam, 1081 HV Amsterdam, The Netherlands; n.vandedonk@amsterdamumc.nl; 3Jerome Lipper Myeloma Center, Department of Medical Oncology, Dana-Farber Cancer Institute, Harvard Medical School, Boston, MA 02215, USA; paul_richardson@dfci.harvard.edu

**Keywords:** CD38, multiple myeloma, daratumumab, antibody, immunotherapy

## Abstract

During a time span of just a few years, the CD38 antibody, daratumumab, has been established as one of the most important new drugs for the treatment of multiple myeloma, both in the relapsed/refractory setting and, more recently, as a first-line treatment. Although much is known about the pleiotropic modes of action of daratumumab, we are still not sure how to use it in an optimal manner. Daratumumab targets CD38 on myeloma cells and a high level of CD38 expression facilitates complement-mediated cytotoxicity (CDC), antibody-dependent cellular cytotoxicity (ADCC) and antibody-dependent cellular phagocytosis (ADCP). Since the expression of CD38 by myeloma cells is downregulated during treatment with daratumumab, it may seem reasonable to introduce a wash-out period and retreat with daratumumab at a later time point when CD38 expression has recovered in order to gain the maximum benefit of daratumumab’s capacity to kill myeloma cells by CDC, ADCC and ADCP. In other aspects, CD38 seems to serve as a survival factor for myeloma cells by facilitating protective myeloma cell–stromal-cell interactions, contributing to the formation of nanotubes that transfer mitochondria from the stromal cells to myeloma cells, boosting myeloma cell proliferation and survival and by generation of immunosuppressive adenosine in the bone marrow microenvironment. In addition, continuous exposure to daratumumab may keep immune suppressor cells at a low level, which boosts the anti-tumor activity of T-cells. In fact, one may speculate if in the early phase of treatment of a myeloma patient, the debulking effects of daratumumab achieved by CDC, ADCC and ADCP are more important while at a later stage, reprogramming of the patient’s own immune system and certain metabolic effects may take over and become more essential. This duality may be reflected by what we often observe when we watch the slope of the M-protein from myeloma patients responding to daratumumab: A rapid initial drop followed by a slow decline of the M-protein during several months or even years. Ongoing and future clinical trials will teach us how to use daratumumab in an optimal way.

The CD38 antibody, daratumumab, has been established as one of the most promising drugs for treatment of multiple myeloma in recent years. It has demonstrated activity as a single agent and in combination with several standard-of-care anti-myeloma drugs both for relapsed/refractory myeloma and in the first-line setting [1,2,3,4,5,6,7] Addition of daratumumab to standard of care anti-myeloma drugs has generally improved the depth of response and PFS globally and across all major subgroups of patients but perhaps without fully compensating for the impact of high-risk cytogenetics. The approved dose and schedule of daratumumab was determined by detailed pharmacokinetic studies conducted during the GEN501 trial, but although most patients probably receive optimal treatment following these guidelines, it is still uncertain if patients with a suboptimal response or resistance to daratumumab could benefit from higher doses or more frequent dosing of Daratumumab. During GEN501, no maximum tolerated dose was found at doses of up to 24 mg/kg. The optimal duration of treatment with daratumumab has not been determined, but responses tend to deepen over time, with more patients becoming minimal residual disease-negative during three years of treatment and perhaps, even longer. Stopping rules for treatment have not been determined, but clinical trials are being planned to see if treatment with daratumumab can be interrupted in patients that have been MRD-negative for two years.

Careful analysis of bone-marrow samples collected during the first clinical trials with daratumumab monotherapy (GEN501 and Sirius) showed that patients with a relatively high expression of CD38 by the myeloma cells had a higher likelihood of achieving a partial response or better, when compared to patients whose tumor cells had lower cell surface expression of CD38 [8]. It was also found that immediately after initiation of treatment with daratumumab, the expression by myeloma cells of CD38 drops to a low level, which remains low for the duration of therapy with daratumumab [8]. This reduction in CD38 cell surface expression occurs both in responding and non-responding patients. 

Selective elimination of myeloma cells with high CD38 expression and survival of myeloma cells with low CD38 expression could potentially explain a reduced expression of CD38, but since the phenomenon is also observed in non-responding patients, this explanation may be less likely. It has been shown that shedding or transfer of daratumumab-CD38 complexes from tumor cells to extracellular fluids (“capping” followed by shedding) or to immune effector cells (“trogocytosis”) may result in reduced levels of CD38 on the tumor cell surface [9,10]. 

At the time of treatment failure and development of progressive disease, there is no further reduction of the expression of CD38 by myeloma cells. This indicates that reduced levels of CD38 expression do not seem to contribute to treatment failure. When treatment with daratumumab is stopped, the myeloma cells will gradually start to re-express higher levels of CD38 [8]. Based on this observation and preclinical findings of better activity of daratumumab against myeloma cells both by complement-mediated cytotoxicity (CDC), antibody-dependent cellular cytotoxicity (ADCC) and antibody-dependent cellular phagocytosis (ADCP) when the level of CD38 expression is high [8,11], it has been suggested that retreatment with daratumumab could be a reasonable strategy after a “wash-out” period of 3–6 months to allow for recovery of CD38 expression. This strategy is now being tested in a clinical trial. However, one may worry if a recovery period could lead to the emergence of resistant clones. It has also been proposed that pharmacological manipulation of the expression of CD38 by myeloma cells may tip the balance in favor of a better response to daratumumab. Both all-trans retinoic acid (ATRA) and panobinostat have been shown to increase the expression of CD38 by myeloma cells, resulting in enhanced lysis of MM cells by daratumumab in vitro and in myeloma mouse models [12,13]. The effect of ATRA in myeloma patients failing to respond to daratumumab or progressing after a prior response to daratumumab is currently being tested in a clinical trial.

A puzzling observation from the GEN501 and Sirius trials is that many patients may continue to respond to daratumumab even when the CD38 expression by myeloma cells is low. The demonstration of an immunomodulatory effect of daratumumab may provide an explanation for the sustained disease control exerted by daratumumab even when the CD38 expression is low [14]. Immunosuppressive regulatory cells of the T-, B- and myeloid lineages are eliminated by daratumumab, causing cytotoxic T-cells to expand in a clonal manner and to become activated. In addition, CD38 is an ectoenzyme that may contribute to the generation of immunosuppressive adenosine and the lower expression of CD38 during treatment with daratumumab may lead to reduced production of adenosine in the bone marrow microenvironment of the myeloma cells and better T-cell mediated immune control of the disease [9,15]. 

CD38 also functions as an adhesion molecule, and its downregulation may lead to altered interactions with the protective bone-marrow microenvironment [16]. 

A recent pre-clinical observation may lend further support to the notion that low expression of CD38 by myeloma cells may in fact be beneficial. Marlein and colleagues demonstrated, first in model systems of acute myeloid leukemia and subsequently, in myeloma, that the cancer cells may interact with bone-marrow stromal cells and form nanotubes that transfer mitochondria from the stromal cells to the cancer cells and provide energy for proliferation and improved survival [17]. These nanotubes are dependent on CD38 for their formation and both genetic knock-down of CD38 and treatment with CD38 antibody interfere with the formation of these nanotubes and with mitochondrial transfer, and thereby, inhibit proliferation and reduce survival of the cancer cells. The various modes of action of daratumumab when used for treatment of myeloma are summarized in Appendix A.

We need to know more about the clinical relevance of the transfer of mitochondria through nanotubes from the stromal cells to myeloma cells as a way of supporting the proliferation and survival of myeloma cells, but it is tempting to speculate that keeping CD38 expression low by continued exposure to daratumumab may inhibit the proliferation of myeloma cells and increase their vulnerability to concomitant anti-myeloma therapy.

Both the inhibition of formation of nanotubes and reduced production of immunosuppressive adenosine may speak in favor of continued treatment with daratumumab without any interruptions but with a change of concomitant anti-myeloma drugs at the time of progressive disease, even and in fact, especially if CD38 expression by the myeloma cells is low. Continued exposure to daratumumab may also help to keep the numbers of immunosuppressive regulatory T-cells, regulatory B-cells, and myeloid-derived suppressor cells at low levels and thereby improve T-cell-mediated tumor control. Preclinical studies of in vitro and in vivo models have shown that genetic ablation or antibody-mediated blockade of CD38 may boost the anti-tumor activity of cytotoxic T-cells directly—a finding that may be clinically relevant and speak in favor of continued, uninterrupted treatment with daratumumab [18].

On the other hand, we know that killing of myeloma cells with daratumumab mediated by CDC, ADCP, or ADCC is more efficient when CD38 expression is high. If we rely primarily on CDC, ADCP, or ADCC for control of myeloma it makes sense to introduce a treatment-free interval, which will allow re-expression of CD38 by the myeloma cells and subsequently, introduce a re-treatment approach where daratumumab is used again in an appropriate new combination. Another approach could be to pharmacologically enhance CD38 expression by treatment with ATRA or panobinostat and thereby improve the efficacy of daratumumab.

At this point in time, we are left with many unanswered questions regarding the optimal way of using daratumumab. Carefully planned clinical trials combined with bench-side work will step-by-step improve our understanding of the extremely complex modes of action when daratumumab is used for treatment of myeloma and guide us on how to optimize the outcome of our treatment.

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
