# Peer review of "Controversy in the Use of CD38 Antibody for Treatment of Myeloma: Is High CD38 Expression Good or Bad?"

_cells, 2020, doi:10.3390/cells9020378_

Round 1

Reviewer 1 Report

The paper by Plesener, van de Donk and Richardson is a nice and brilliant representation of issues related to the use of therapeutic antibodies and the analysis of the functions featured by the target molecule. The manuscript is clear, schematical and useful for clinicians and basic scientists working in the field. My only comment is related to the ability of Daratumumab to influence the ectoenzymatic functions of the target molecule. If possible, the authors could include a general reference dealing with this issue:

CD38 in Adenosinergic Pathways and Metabolic Re-programming in Human Multiple Myeloma Cells: In-tandem Insights From Basic Science to Therapy.

Horenstein AL, Bracci C, Morandi F, Malavasi F.

Front Immunol. 2019 Apr 24;10:760. doi: 10.3389/fimmu.2019.00760. eCollection 2019. Review.

PMID: 31068926  

Author Response

The proposed reference has been added as ref 15

Reviewer 2 Report

This is a well written paper which describes the controversy in the use of daratumumab and the role of CD38 expression. I just have a few minor comments:

1) There is no conclusion in the abstract.  

2) The authors could discuss  that "introducing a wash-out period and retreating with daratumumab at a later time point when CD38 expression has recovered" could potentially lead to selection and growth of resistant tumor cells.

3) They could also discuss the role of cytogenetics, e.g. should high risk markers such as del(17p) be considered in this context?

Author Response

Please see lines 35-36, 43-46 and 82-83 of the attachment

Reviewer 3 Report

This is an exciting, timely and well written comment to aspects of how we currently think (or belive) daratumumab is working in Treatment of myeloma patients. The manuscript is easy to read, entertaining. and will be stimulating people working in the field.

This reviewer has no major issues. As minor comments, the authors are invited to speculate on dosing issues (are we giving enough ? is pharmacologic monitoring useful?) related to the use of daratumumab, on the currently used intervals (reasonable?), and should be more precise on the duration of Treatment with dara (what does on-off/retreatment mean exactely? possibly, a Figure/schedule on this aspect may be helpful ?).

Author Response

Please see lines 46-55 in the attachment
